# Oxidation of Gaseous Elemental Mercury in Acidified Water: Evaluation of Possible Sinking Pathway of Atmospheric Gaseous Mercury in Acid Cloud, Fog, and Rain Droplets

Satoshi Irei

Department of Environment and Public Health, National Institute for Minamata Disease, 4058-18 Hama, Minamata, Kumamoto 867-0008, Japan; satoshi.irei@gmail.com; Tel.: +81-966-63-3111 (ext. 760)

**Abstract:** This is the first report investigating the transformation of gaseous elemental mercury (GEM), the major form of airborne mercury, into oxidized mercury in bulk liquid, a possible sinking pathway of atmospheric GEM in clouds, fog, rain droplets and ocean spray. A 100–150 ng m$^{-3}$ GEM standard gas, a 50–150 times higher concentration than the typical atmospheric concentration, was introduced into a 2.5 L rectangular glass vessel, at the bottom of which a 0.5 L uptake solution of pure water (pH 6–7), weakly acidified pure water with sulfuric or nitric acid (pH 3.2–3.6) or seawater (pH 8) was resting. The standard gas was introduced into the space above the solution in the vessel at the rate of 0.82 L min$^{-1}$ and exited from the opposite end of the vessel, which was open to the room's pressure. After exposing the solution to the gas for 0.5–4 h, a portion of the uptake solution was sampled, and the dissolved elemental mercury ($Hg^0_{aq}$) and dissolved oxidized mercury ($Hg^{2+}_{aq}$) in the solution were analyzed by the conventional trapping method, followed by cold vapor atomic fluorescent spectrometer measurements. The results showed that the quantities of total dissolved mercury ($THg_{aq} = Hg^0_{aq} + Hg^{2+}_{aq}$) in the pure water and seawater were compatible, but those were slightly lower than the equilibrated $Hg^0_{aq}$ concentrations estimated from Henry's law, suggesting non-equilibrium throughout the whole solution. In contrast, the quantity of $Hg^{2+}_{aq}$ and $THg_{aq}$ in the acidified pure water with sulfuric acid was significantly enhanced. Over the 4 h exposure, the $THg_{aq}$ concentrations were two times higher than the equilibrated $Hg^0_{aq}$ concentration. This was due to the slow oxidation reaction of $Hg^0_{aq}$ by the sulfuric acid in the bulk phase. Using the collision rate of GEM with the surface of the solution and the observed uptake, the estimated uptake coefficient of GEM by this uptake was $(5.5 \pm 1.6) \times 10^{-6}$. Under the typical atmospheric concentration, this magnitude results in an atmospheric lifetime of 4970 years, negligibly small compared with other atmospheric oxidation processes.

**Keywords:** gaseous elemental mercury; dissolved elemental mercury; dissolved oxidized mercury; heterogeneous uptake; acid cloud and fog droplet; acid rain; oxidation of mercury

## 1. Introduction

Mercury is a toxic metal and has a unique characteristic among metal elements: it evaporates under room temperature and pressure conditions. Due to this, mercury easily enters the natural environment though the atmosphere and spreads all over the globe. Human activity is recognized as one of the major contributors for mercury found in the natural environment, accounting for 24% of the total mercury emission in the atmosphere [1]. Consequently, the amount of mercury contained in wildlife and human bodies has been increasing since the industrial era [2]. Therefore, the United Nations has been implementing international regulations on the use of mercury, such as the Minamata Convention on Mercury, to reduce the level of global pollution.

Generally, atmospheric mercury is categorized into three species: gaseous elemental mercury (GEM), gaseous oxidized mercury (GOM) and particulate boundary mercury [3,4]. It has been recognized that the majority of atmospheric mercury exists in the form of

GEM [5], comprising 81–96% of atmospheric mercury [6], which is sparingly reactive and soluble to water [7–9]. Once GEM is oxidized and transformed to GOM, it is adsorbed and absorbed by either atmospheric droplets and particulate matter or aerosols and deposits on the Earth's surface, consequently being efficiently scavenged from the atmosphere. Thus, the oxidation reaction of GEM in the gas phase is a key step to determining the fate of atmospheric GEM [10]. Based on the rate laws of GEM oxidation reactions with the major atmospheric oxidants, such as OH radicals, ozone and bromine radicals [11], the estimated atmospheric lifetime of GEM is half a year to one year [12]. However, there are unidentified processes that may significantly affect the fate of atmospheric mercury [13]. For example, Holmes et al. [14] reported the comparison of their model studies with the results from real observations for atmospheric mercury concentrations, and they found a discrepancy between the model and the observations; the actual concentration decrease of atmospheric mercury was faster than the estimated rate. They concluded that this could imply an unidentified chemical reaction or reactions involved, possibly the reaction with the chlorine radical in the marine atmosphere. Among the atmospheric mercury oxidations thought to occur in the natural environment [13,15], there are more unevaluated processes for GEM, and the reactive heterogeneous uptake by acid water is one of them.

To date, the heterogeneous uptake of gaseous chemical species by aerosols, cloud droplets and ocean spray has been reported [16]. There has been a report of oxidation reaction of mercury in liquid phase with ozone [17], but none of the reports have studied the uptake of GEM by acid water. This is because mercury is less reactive than hydronium ions in the activity series, and the oxidation reactions are often slow. This is also because the equilibrium assumption between the atmospheric GEM and water (clouds, fog, rain droplets and sea spay), that being the solubility of GEM, may deliver a reasonable estimation for the amount of mercury directly dissolved in atmospheric water. Contradictory to the mercury reaction with acids, it is known that some inorganic acids, such as sulfuric acid and a mixture of nitric and hydrochloric acids, oxidize and dissolve metal mercury [18–20]. To the best of our knowledge, the acidity of atmospheric water has not been considered in the calculation of the mercury cycle in the natural environment yet. It is known that, like acid rain, atmospheric water becomes acidic when secondary pollutants, such as sulfuric and nitric acids, are absorbed. In the East Asian region, a low pH, such a pH of 2.9 in the cloud water [21] and a pH between 3 and 4 in the rainwater [22,23], has been frequently observed due to the large anthropogenic emissions of precursor gases for sulfuric and nitric acids. When such acidic droplets meet with GEM, it is possible that the uptake of GEM can be enhanced. The observed high mercury concentrations in the low pH (<4) cloud water collected at Mt. Bamboo in Taiwan may be due to the oxidation of GEM in the acidic cloud water. In this research, we conducted laboratory experiments for the heterogeneous uptake of GEM on bulk solutions under room lighting and near the standard temperature (293–299 K) and pressure (1.0–1.6 Pa above the room's air pressure) conditions to evaluate if the heterogeneous uptake was significant.

Approximately 100–150 ng m$^{-3}$ of a GEM standard gas was introduced into a 2.5 L rectangular glass vessel, at the bottom of which a 500 mL uptake solution of pure water, acidified water (with sulfuric or nitric acid, typical anthropogenic acidic pollutants) or natural seawater rested. After exposure to the standard gas for a given period of time, the solutions—sampled and dissolved mercury species, dissolved elemental mercury ($Hg^0_{aq}$) and dissolved oxidized mercury ($Hg^{2+}_{aq}$)—were separately extracted using the conventional purging and trapping method with only an inertial gas or together with a spiking $SnCl_2$ solution. Then, they were quantitatively analyzed by a fluorescent spectrometer. The acquired data were then evaluated to see if this process was significant in a real atmosphere.

## 2. Materials and Method

### 2.1. Uptake Experiment

Uptake experiments of GEM by bulk solutions were conducted using a rectangular glass vessel (50 cm length × 10 cm width × 5 cm depth, COSMOS VID, Fukuoka, Japan)

under room lighting, temperature and pressure conditions. A three-eighths inch o.d. glass tubing was fused to the both ends of the vessel (Figure 1) so that a GEM standard gas could flow into and out of the vessel via the three-eighths inch tubing. Prior to use, the vessel was thoroughly cleaned in the following manner to reduce the background of mercury: 30% of inversed aqua regia was filled fully and left for at least 2 days, followed by rinsing with tap water at least 6 times and with Milli-Q water 6 times, and then drying with mercury-free air, which was prepared using compressed room air (super oil free BEBICON 0.4LE-8SB, Hitachi Industrial Equipment Systems Co., Ltd., Tokyo, Japan), a dryer stuffed with silica gel (Kanto Chemical Co., Inc., Tokyo, Japan) and a mercury trap stuffed with activated charcoal (Hokuetsu MA-HG, Ajinomoto Fine Techno Co., Inc., Yokohama, Japan). The background mercury in the mercury-free air, analyzed by the method utilizing conventional gold trapping and a cold vapor atomic fluorescent spectrometer or CV-AFS (WA-5F, Nippon Instruments Corp.), was 7 pg m$^{-3}$ or below. This whole cleaning procedure was repeated again using 20% inversed aqua regia and rinsing with tap and Milli-Q water at least 15 times prior to use for the uptake experiments.

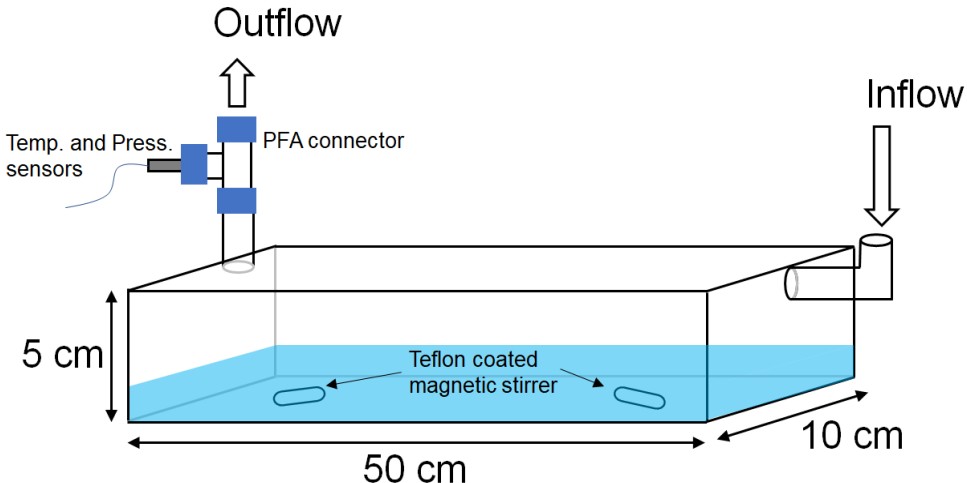

**Figure 1.** Schematic of the uptake experiment of gaseous elemental mercury (GEM) by a bulk solution.

The uptake solutions used were either fresh pure water (ultrapure water, Kanto Chemical Co. Inc.), acidified water prepared by mixing 600 mL of fresh pure water with a drop of sulfuric (ACS grade, Nacalai Tesque Inc., Kyoto, Japan) or nitric acid (ACS grade, Kanto Chemical Co. Inc.) right before the experiments or sea water sampled from the coast of Akune city, Kagoshima, Japan (31.6° N, 130.1° E), which is located northeast of the East China Sea. The sea water was filtered in advance with 47 mm PTFE filters (Omnipore, 0.45 mm PTFE membrane, Merck Millipore Ltd., Watford, UK). It should be noted that the typical volumes of an acid droplet, according to the mass measurements using a mass balancer (XPE502, Mettler-Toledo GmbH, Greifensee, Switzerland), are $9 \pm 1$ µL and $13 \pm 1$ µL for sulfuric and nitric acid, respectively. The background concentrations of $Hg^0_{aq}$ and $Hg^{2+}_{aq}$ were $23 \pm 7$ pg L$^{-1}$ and $106 \pm 66$ pg L$^{-1}$ for the ultrapure water ($n = 6$, average $\pm$ SD (standard deviation)) and 60 pg L$^{-1}$ and 81 pg L$^{-1}$ ($n = 1$) for the non-purged seawater, respectively. The seawater was purged for two days with mercury-free air for the reduction of background mercury prior to use for the uptake experiments. However, the background concentrations of $Hg^0_{aq}$ and $Hg^{2+}_{aq}$ in the purged seawater remained the same. The pH of the uptake solution was measured before and after the GEM exposure using a pH meter (D-71, HORIBA Advanced Techno Co. Ltd., Kyoto, Japan), which was calibrated every time it was used. The glass vessel, in which two PTFE-coated magnet stirrers were placed, was placed on a weighing balancer (KERN and SOHN GmbH, Balingen, Germany), and 500 g of the uptake solution was introduced gently into the glass vessel using a polypropylene long nose funnel (FWC220, AS ONE Co., Ltd., Osaka, Japan) so that the solution was filled from the bottom of the vessel. The vessel with the uptake

solution was weighed before and after the uptake experiments to check the evaporation loss of the uptake solution during the experiments. The vessel was kept horizontal during the exposure. A relative pressure sensor (GC30, Nagano Keiki Co., Tokyo, Japan) and a temperature and relative humidity sensor (TA502RW, Toplas Engineering Co. Ltd., Tokyo, Japan) were set at the outlet of the vessel to monitor the pressure and temperature of the standard gas outflow. The analog signal output from these sensors were logged via a PC data logger (NR500 and NRTH08, KEYENCE, Osaka, Japan).

The standard GEM gas was generated using a permeation device (PD-1B-2, GASTEC, Ayase, Japan) with liquid mercury enclosed by a permeation tube (VICI AG International, Schenkon, Switzerland). The temperature of the device was set to 323 K all the time, and the flow of mercury-free air into the permeating chamber was set to 0.82 L min$^{-1}$. No split flow was set; therefore, the inflow into the permeation device corresponded to the outflow from the device. The concentration of GEM was determined by sampling and measuring the GEM standard gas with a conventional gold trap (Nippon Instruments Corp., Osaka, Japan) and a CV-AFS, which was calibrated with saturated GEM under a given temperature (MB-1, Nippon Instruments Corp.). The GEM sampling was done at the inlet of the glass vessel because pilot tests showed almost no difference in the GEM concentrations between the inlet and outlet of the glass vessel during exposure. The GEM sampling was done at least before and after exposure in each experiment. The GEM standard gas from the permeation device was continuously introduced from one end of the glass vessel and flowed out from another end, which was open to the air. During exposure, the magnetic stirrers were ceased. For safety reasons, all uptake experiments were carried out in a fume hood. After exposing the solution to the GEM standard gas for a given period of time, the introduction of the standard gas was stopped, and the solution was gently stirred using the magnetic stirrers for 3–5 min. Approximately 290 mL of the homogenized uptake solution was gently siphoned from the bottom of the vessel into a 375 mL PFA impinger (Savillex, Minneapolis, MN, USA) using an approximately 60 cm long, 6 mm i.d. × 8 mm o.d. PFA tubing (Yodoflon, 8 mm o.d.× 6 mm i.d., Yodogawa Hu-Tech Co., Ltd., Osaka, Japan). The mass of the uptake solution sample in the impinger was weighed using a balancer (ML4002T, Mettler-Toledo GmbH). The analysis of mercury in the uptake solutions described in the following subsection was carried out as soon as the solutions were decanted in the impinger.

## 2.2. Extraction for Dissolved Gaseous and Oxidized Mercury

To better understand the process of GEM uptake, it is critical to separately extract $Hg^0_{aq}$ and $Hg^{2+}_{aq}$ in a solution. We used a conventional purging method with an inertial gas, nitrogen, to extract $Hg^0_{aq}$ and $Hg^{2+}_{aq}$ separately from the same solution [24,25]. First, a sample solution in a 375 mL PFA impinger was purged with mercury-free nitrogen (nitrogen gas passed through a gold mercury trap) via a glass bubbler (G-3, COSMOS VID) at a flow rate of 0.5 L min$^{-1}$ for a given period of time. The $Hg^0_{aq}$ in the solution was purged out and captured by a conventional gold trap (Nippon Instruments Corp.) through a soda lime that dried the outflow gas. After this extraction, a 50 cm long piece of one-quarter inch o.d. PFA tubing (Tombo, 9003-PFA, NICHIAS Corp, Tokyo, Japan) was connected to the impinger. Then, 5 mL of a 20% (*w/w*) tin chloride (99% purity, Kanto Chemical Co., Inc.) solution in a 10% hydrochloric acid (Kanto Chemical Co., Inc.) solution was spiked to the residual solution in the impinger via the PFA tubing and purged with mercury-free nitrogen. The $Hg^{2+}_{aq}$ in the solution was reduced to $Hg^0_{aq}$ by $Sn^{2+}$, purged out from the solution and captured in the same manner as in the extraction of $Hg^0_{aq}$. The trapped mercury was quantitatively analyzed by the CV-AFS. It should be noted that after the extraction, the glassware and containers needed to be thoroughly cleaned for the next extraction. In particular, extra care was needed not only to remove residual mercury, but to quench the residual $Sn^{2+}$, which is a fast-reducing reagent of $Hg^{2+}_{aq}$, but it also turned out to be a stubborn contaminant, interfering with the analysis of $Hg^0_{aq}$ and $Hg^{2+}_{aq}$ in a solution. Our cleaning procedure was as follows: the impingers and gas bubblers were

soaked in a 30% (*v/v*) inversed aqua regia mixture for a week, rinsed thoroughly with tap water 15 times, followed by rinsing with Milli-Q water 15 times and, finally, drying in a drying oven set to 353 K (DG400, Yamato Scientific Co. Ltd., Tokyo, Japan) for at least two days. This cleaning procedure was performed twice prior to use for the next extraction.

For optimization of the purging time, we used an SRM 8610 $Hg^{2+}$ standard solution (NIST, Gaithersburg, MD, USA). It should be noted that the reference mercury concentration of this standard material was analyzed and provided by NIST, but the value was not certified. In the impinger, 145–310 pg of $Hg^{2+}$ was spiked to 290 mL pure water. Then, $Hg^{2+}_{aq}$ was extracted in the same manner described above with a variety of purging times. This test confirmed the unfavorable reduction of $Hg^{2+}_{aq}$ by just purging the solution with the mercury-fee air, followed by the recovery yields of $Hg^{2+}_{aq}$ using the tin chloride reduction method. Unfortunately, we could not directly evaluate the recovery yields for $Hg^0_{aq}$ because there was no standard solution available for $Hg^0_{aq}$. However, we assumed that the optimum purging time for $Hg^0_{aq}$ corresponded to that of $Hg^{2+}_{aq}$. This is because $Sn^{2+}$ very effectively and quickly converts $Hg^{2+}$ to $Hg^0_{aq}$ and, therefore, the extraction or purging time for $Hg^{2+}_{aq}$, optimized using the SRM 8610, was dependent on the extraction efficiency of $Hg^0_{aq}$ reduced from $Hg^{2+}_{aq}$ by $Sn^{2+}$.

For the purpose of quality control, a number of sample blanks were taken in the following manner: A quantity of 300 mL of the ultrapure water or filtered and purged seawater was poured into the cleaned 2.5 L glass vessel, and the vessel was thoroughly rinsed. Then, 290 mL of the rinsed ultrapure water was decanted into the impinger, and the $Hg^0_{aq}$ and $Hg^{2+}_{aq}$ in the sample blanks were analyzed in the same manner as described previously. The obtained blank values were deducted from the results of the uptake experiments.

## 3. Results and Discussion

### 3.1. Detection Limits and Sample Blank Values

The detection limit of our CV-AFS was 0.1 pg. The average sample blank values $\pm$ SDs for the $Hg^0_{aq}$ and $Hg^{2+}_{aq}$ were $20 \pm 10$ pg $L^{-1}$ and $100 \pm 50$ pg $L^{-1}$ for the ultrapure water (*n* = 33) and $31 \pm 9$ pg $L^{-1}$ and $109 \pm 46$ pg $L^{-1}$ for the seawater (*n* = 4), respectively. The blank values for the two different types of water were comparable and significantly lower than the concentrations of mercury found in the uptake solutions (presented and discussed below) but substantially higher than the amount of background mercury in the purging gas. This may imply that their source or sources were the same and possibly that the thoroughly cleaned impingers and bubblers may have still been contaminated. Based on the SDs, the detection limits for $Hg^0_{aq}$ and $Hg^{2+}_{aq}$ in a solution were approximately 30 pg $L^{-1}$ and 150 pg $L^{-1}$ or 9 pg and 44 pg, respectively.

### 3.2. Extraction Test

Extraction tests with SRM 8610 spikes of approximately 150 pg and 308 pg with various extraction times showed recovery yields from 64% to 102%, depending on the purging duration (Table 1). For the 150 pg spikes corresponding to 517 pg $L^{-1}$, the highest recovery yields were 86% with the purging time of 120 min while, for the 308 pg spikes (corresponding to 1062 pg $L^{-1}$), the recovery yields reached 102% with the purging time of 120 min. The high recovery yields suggest that the photochemical reduction of $Hg^{2+}$ did not take place under our experimental conditions. It should be noted that some amount of $Hg^0_{aq}$ (2–12% mercury, relative to the amount of $Hg^{2+}_{aq}$) was always observed in the blank corrected results of the $Hg^{2+}_{aq}$ spike tests, implying that up to 12% of the $Hg^{2+}_{aq}$ may have been unfavorably converted to $Hg^0_{aq}$ when purged with the mercury-free nitrogen. This bias was not corrected in the uptake data discussed in the following subsection. It should also be noted that poorer recovery yields for the 150 pg spike tests were likely due to the blank subtraction (i.e., smaller sample size and larger influence from the blank subtraction). Based on these results, the 120 min extraction time was chosen for our analysis. As shown later, all the measured masses were similar to or smaller than the 150 pg range. Therefore,

corrections for the recovery yield (measured mass divided by 0.86, obtained from the recovery yield test) were made.

**Table 1.** Results of the extraction test with 150 pg or 308 pg SRM 8610 spiking in 290 mL pure water.

| Spiked $Hg^{2+}$ | § Purging Time | $n$ | † $Hg^0_{aq}$ | † $Hg^{2+}_{aq}$ | † $Hg^{2+}_{aq}$ Recovery Yield |
|---|---|---|---|---|---|
| (pg) | (min) | | (pg $L^{-1}$) | (pg $L^{-1}$) | (%) |
| 150 | 15 | 5 | $23 \pm 5$ | $328 \pm 46$ | $64 \pm 14$ |
| 150 | 30 | 5 | $30 \pm 7$ | $348 \pm 38$ | $68 \pm 11$ |
| 150 | 60 | 4 | $12 \pm 2$ | $375 \pm 29$ | $73 \pm 8$ |
| 150 | 90 | 3 | $42 \pm 16$ | $343 \pm 27$ | $67 \pm 8$ |
| 150 | 120 | 2 | $27 \pm 10$ | $446 \pm 25$ | $86 \pm 6$ |
| 308 | 60 | 3 | $16 \pm 5$ | $952 \pm 26$ | $90 \pm 3$ |
| 308 | 90 | 3 | $40 \pm 4$ | $1000 \pm 20$ | $94 \pm 2$ |
| 308 | 120 | 3 | $32 \pm 5$ | $1088 \pm 24$ | $102 \pm 2$ |

§ The time individually spent for $Hg^0_{aq}$ and $Hg^{2+}_{aq}$ extraction. † Average $\pm$ standard deviation (SD) of the blank corrected values.

*3.3. Uptake Experiment*

The overall results of the uptake experiments are summarized in Table 2. During the period of each exposure experiment, the concentration of GEM produced by the permeator was stable. Because of the air conditioning environment, the temperature during each experiment was also stable, fluctuating within $\pm$ 0.6 K. An example of time series observations for temperature and pressure are shown in Figure 2. The inflow was kept constant, and the pressure difference between the inside and outside of the glass vessel was smaller than 1.6 Pa during exposure. The weight measurements of the vessel with the solution before and after the standard gas exposure showed a decrease of up to 5 g over the 4 h exposure and depended on the exposure period. Because evaporation influenced the volume of the uptake solution by only approximately 1%, we did not make any correction to the volume for the determination of $Hg^0_{aq}$ and $Hg^{2+}_{aq}$ concentrations. Even though it is not shown here, occasional temperature measurements of the exposed solution by a thermometer showed the differences in temperature between the solution and room air were approximately 1 K (lower for the solutions). It should be noted that the concentrations for $Hg^0_{aq}$ and $Hg^{2+}_{aq}$ discussed hereafter were blank corrected, and the pH measurements of the uptake solution after exposure demonstrated no change in pH.

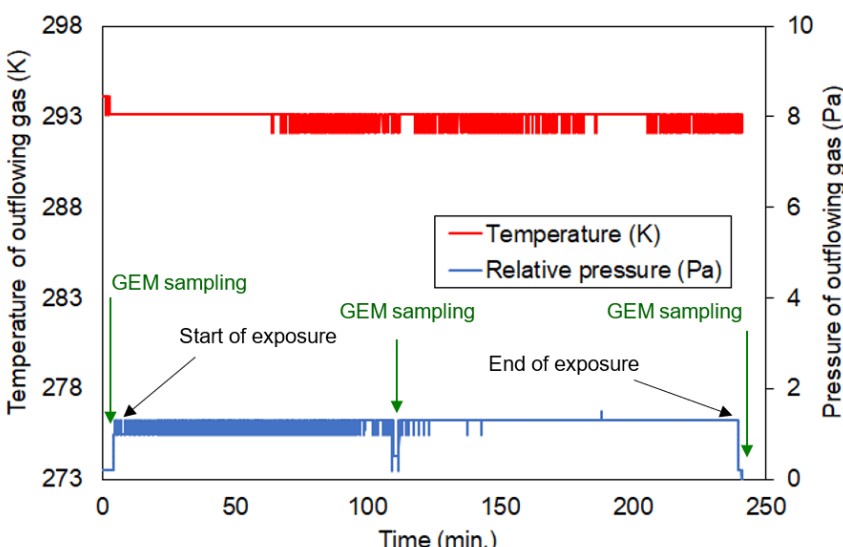

**Figure 2.** Time series plot of the temperature (T) and relative pressure (P) of the outflowing GEM standard gas from the glass vessel during the 4 h exposure experiment with a dilute nitric acid solution.

**Table 2.** Results of GEM uptake experiments by variety of bulk solutions.

| Sample | Exposure Duration | Mass of Uptake Solution | pH of Uptake Solution | § GEM Concentration | § Temperature | § Relative Pressure | ‡ $Hg^0{}_{aq}$ | ‡ $Hg^{2+}{}_{aq}$ |
|---|---|---|---|---|---|---|---|---|
| | (min) | (g) | | (ng m$^{-3}$) | (K) | (Pa) | (pg L$^{-1}$) | (pg L$^{-1}$) |
| Pure water | 60 | 499.7 | † 6.1–6.5 | 153 ± 9 | 299.2 ± 0.2 | 0.70 ± 0.02 | 13.7 | 308 |
| Pure water | 153 | 500 | † 6.1–6.5 | 124 ± 3 | 298.4 ± 0.4 | 1.29 ± 0.06 | 121.2 | 170.9 |
| Pure water | 181 | 500.6 | † 6.1–6.5 | 131.2 ± 0.8 | 299.1 ± 0.3 | 1.2 ± 0.1 | 279.5 | 268.7 |
| Pure water | 241 | 499.7 | † 6.1–6.5 | 116 ± 3 | 297.1 ± 0.3 | 1.1 ± 0.2 | 184.4 | 263 |
| dilute $H_2SO_4$ | 122 | 500.3 | 3.3 | 130 ± 10 | 295.2 ± 0.2 | 1.02 ± 0.08 | 58.8 | 614.6 |
| dilute $H_2SO_4$ | 60 | 502.2 | 3.24 | 105 ± 6 | 293.4 ± 0.4 | 1.6 ± 0.1 | 50.4 | 325.2 |
| dilute $H_2SO_4$ | 238 | 500.3 | 3.25 | 110 ± 3 | 297.3 ± 0.3 | 1.32 ± 0.06 | 83.7 | 742.2 |
| dilute $H_2SO_4$ | 30 | 500.4 | 3.3 | 102 ± 8 | 292.6 ± 0.5 | 1.30 ± 0.02 | 30.7 | 157.6 |
| dilute $H_2SO_4$ | 180 | 500.1 | 3.2 | 112 ± 2 | 294.2 ± 0.5 | 1.29 ± 0.06 | 48 | 918.4 |
| dilute $H_2SO_4$ | 240 | 500 | 3.27 | 110 ± 3 | 295.6 ± 0.6 | 1.30 ± 0.02 | 16.4 | 798.1 |
| dilute $HNO_3$ | 120 | 499.6 | 3.59 | 108.1 ± 0.5 | 294.15 ± 0.01 | 1.1 ± 0.2 | 156.9 | 164.6 |
| dilute $HNO_3$ | 180 | 500.5 | 3.64 | 114.6 ± 0.9 | 295.7 ± 0.5 | 1.1 ± 0.1 | 238 | 198.7 |
| dilute $HNO_3$ | 236 | 500.7 | 3.5 | 106 ± 2 | 293.1 ± 0.3 | 1.3 ± 0.1 | 117.7 | 379.8 |
| dilute $HNO_3$ | 60 | 500.2 | 3.6 | 109.4 ± 0.8 | 295.2 ± 0.1 | 1.2 ± 0.1 | 151.8 | 220 |
| seawater | 239 | 500 | 8.11 | 113.3 ± 0.6 | 296.4 ± 0.4 | 1.2 ± 0.1 | 194.8 | 52.6 |
| seawater | 190 | 501.4 | 7.92 | 107 ± 3 | 294.2 ± 0.1 | 1.1 ± 0.1 | 223.5 | * LDL |
| seawater | 120 | 489.9 | 7.97 | 101 ± 3 | 295.0 ± 0.5 | 1.30 ± 0.01 | 262.7 | 65.5 |
| seawater | 60 | 499.8 | 7.95 | 115.4 ± 0.2 | 295.6 ± 0.5 | 1.03 ± 0.09 | 219.2 | 70.9 |

§ Values shown are the average ± SD. † Values are the range typically observed for blank samples of pure water. ‡ Recovery yield correction (division by 0.86) was made for the concentrations of $Hg^0{}_{aq}$ and $Hg^{2+}{}_{aq}$ lower than 517 pg L$^{-1}$, corresponding to 150 pg in 290 mL. * LDL = lower than the detection limit.

The results of the uptake experiments on GEM by pure water, seawater and acidified pure water with sulfuric or nitric acid showed that the $Hg^0_{aq}$ concentrations in the uptake solutions were lower than the amount of $Hg^0_{aq}$ estimated for the equilibrated solution with the GEM standard gas, according to the Henry's law (Figure 3). In the estimation of the equilibrated concentration, we used the average Henry's constants at 293 K and 298 K provided in the reference data report [7], which are approximately 394 bar and 476 bar, respectively, together with the quantity of the exposed solutions, GEM concentrations and temperature observed during the experiments (Table 2). The regression plots for all solutions demonstrated no significant time dependency. The $Hg^0_{aq}$ concentrations in the pure water, acidified water with nitric acid and seawater were, on average ($\pm$SD), 150 $\pm$ 112 pg L$^{-1}$, 166 $\pm$ 51 pg L$^{-1}$ and 225 $\pm$ 41 pg L$^{-1}$, respectively. These average $Hg^0_{aq}$ concentrations were at a comparable level, but nearly one-half to one-third that of the equilibrated $Hg^0_{aq}$ concentrations. This is most likely explained by the uptake solutions reaching the surface of the uptake solution, but not reaching equilibrium throughout the whole solution due to the slow diffusion of $Hg^0_{aq}$. Our experiments exposed the standard gas in constant concentrations of GEM to the surface of the uptake solutions. Thus, it would take some time to reach equilibrium. On the other hand, many methods in the literature used for the determination of equilibrated concentrations bubbled standard GEM gas in solutions, and the physical mixing method would bring equilibrium rapidly. Meanwhile, the average $Hg^0_{aq}$ concentrations in the acidified solution with sulfuric acid (48 $\pm$ 23 pg L$^{-1}$) were lower than those in the other uptake solutions. This difference can be explained by the chemical reaction of $Hg^0_{aq}$ with sulfuric acid during the two-hour extraction of $Hg^0_{aq}$. Because $Hg^0_{aq}$, hydronium and sulfate ions coexisted in the solution even after GEM exposure, the $Hg^0_{aq}$ was likely converted to $Hg^{2+}_{aq}$ during $Hg^0_{aq}$ extraction.

The $Hg^{2+}_{aq}$ in the uptake solutions showed interesting trends (Figure 4). The $Hg^{2+}_{aq}$ concentrations in the seawater were lower than the $Hg^0_{aq}$, while the $Hg^{2+}_{aq}$ concentrations in the other solutions were higher than the $Hg^0_{aq}$. The $Hg^{2+}_{aq}$ concentrations in the pure water (pH 6.1–6.5), acidified water with nitric acid (pH 3.5–3.6) and seawater (pH 7.9–8.1) showed almost no time-dependent change as a two uncertainty range was considered for the slope, and the $Hg^{2+}_{aq}$ concentrations in the acidified water with sulfuric acid (pH 3.3) increased significantly as a function of time up to 1068 pg L$^{-1}$, exceeding the upper limit of the equilibrated $Hg^0_{aq}$ concentration. The series of results suggests the following: (1) the constrained oxidation of $Hg^0_{aq}$ in seawater was likely associated with a high pH, rather than the high ionic strength, while some extent of the oxidation reaction of the $Hg^0_{aq}$ took place in the pure water and acidified water with nitric acid, possibly with dissolved oxygen; (2) the oxidation reaction of $Hg^0_{aq}$ in the acidified solution with sulfuric acid was associated not only with hydronium ions, but with sulfate ions. The first finding was based on the fact that the $Hg^0_{aq}$ concentrations between all solutions were comparable, implying a low diffusivity of $Hg^0_{aq}$ in seawater, due to the other dissolved ions not significantly lowering the GEM uptake under our experimental conditions, and the high-pH seawater containing dissolved oxygen did not oxidize the $Hg^0_{aq}$ as much as the other solutions did. This explanation is also consistent with our general knowledge that $Hg^{2+}_{aq}$ is stable in an acidic solution; that is, hydronium ions are likely a key player in the redox reactions of $Hg^0_{aq}$, but it still seems to require other molecules to oxidize $Hg^0_{aq}$. This is also consistent with our general knowledge that mercury is a less effective reducing element than hydrogen in the activity series. The second finding is also explained by this, and the oxidation is specific to sulfuric acid. The oxidation of mercury with sulfuric acid, which contradicts to the activity series, has been studied for a long time. The overall oxidation mechanism has been postulated as follows [18]:

$$Hg^0 + 2H_2SO_4 = HgSO_4 + SO_2 + 2H_2O$$

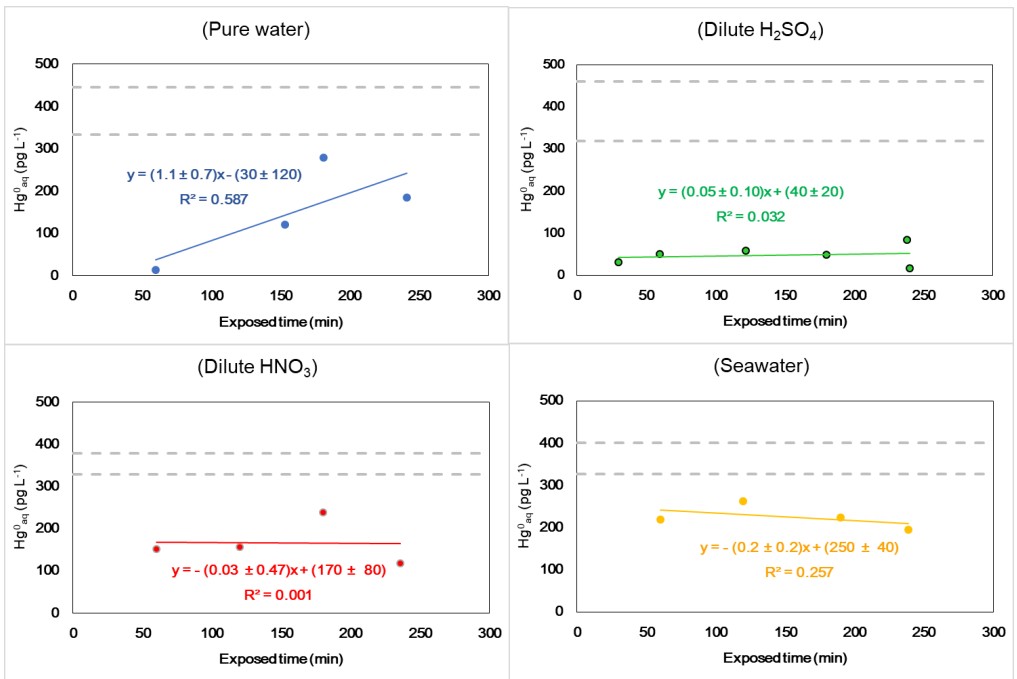

**Figure 3.** Concentration of dissolved elemental mercury ($Hg^0_{aq}$) in the variety of uptake solutions as a function of the exposure time to the standard gas of 101–153 ng m$^{-3}$ gaseous elemental mercury. The gray horizontal lines shown in each plot are for the lowest and highest equilibrated concentrations of $Hg^0_{aq}$, according to Henry's law, estimated with the use of the temperature and GEM concentrations provided in Table 2 and Henry's constants in the literature (7). Linear regressions are also shown in the figures, and the uncertainties shown for each regression are standard errors.

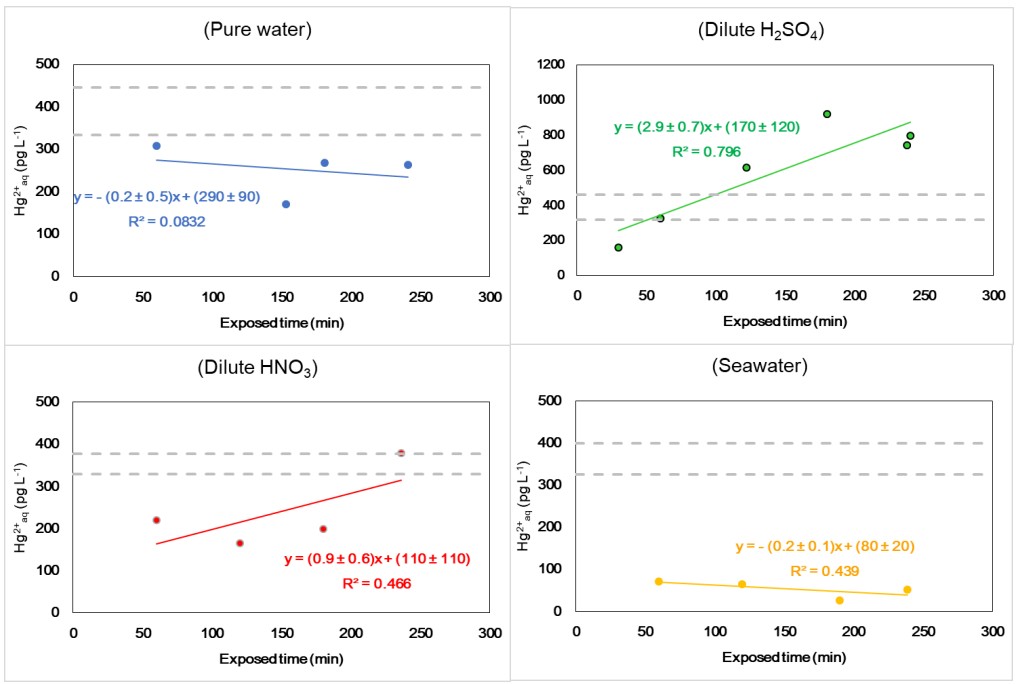

**Figure 4.** Concentration of dissolved oxidized mercury ($Hg^{2+}_{aq}$) in a variety of uptake solutions as a function of the exposure time to the standard gas of 101–153 ng m$^{-3}$ gaseous elemental mercury. The gray horizontal lines shown in each plot are for the lowest and highest equilibrated concentrations of $Hg^0_{aq}$, according to Henry's law, estimated through the use of the temperature and GEM concentrations provided in Table 2 and Henry's constants in the literature [7] The linear regressions are also shown in the f igures, and the uncertainties shown for each regression are standard errors. The lowest $Hg^{2+}_{aq}$ concentration data point in the seawater plot is below the detection limit.

Morris et al. [19] studied the oxidation mechanism of mercury with sulfuric acid. They confirmed the enhancement of GEM uptake by activated carbon coated with sulfuric acid. However, their free energy calculation results showed that the reaction above is not spontaneous. To the best of our knowledge, the oxidation mechanism of $Hg^0_{aq}$ by sulfuric acid (particularly in a dilute sulfuric acid solution) has not been elucidated yet.

The transport processes of GEM in the bulk liquid phase are depicted in Figure 5. A similar illustration with different gaseous reactants has been presented in many publications [16]. Under our experimental conditions, the GEM concentration change from before exposure to the solution to after exposure was very small and insignificant. To have an idea of whether the GEM already reached equilibrium in the four hours of exposure, the concentration of $Hg^0_{aq}$ diffused at the bottom of the solution was estimated. According to Kuss, J. [9], the diffusion coefficient for $Hg^0_{aq}$ in fresh water at 298 K is $1.7 \times 10^{-5}$ cm$^2$ s$^{-1}$. Using this value, together with an arbitrary concentration of $Hg^0_{aq}$ at the surface and a surface area of 500 cm$^2$, only 9% of the $Hg^0_{aq}$ reached to the bottom of the solution, and it took nearly 24 h to have a uniform concentration throughout the solution, suggesting the condition was not at equilibrium in our experiments. The fact that the concentrations of $Hg^{2+}_{aq}$ exceeded the equilibrium in the dilute sulfuric acid solution was likely due to the reactive uptake.

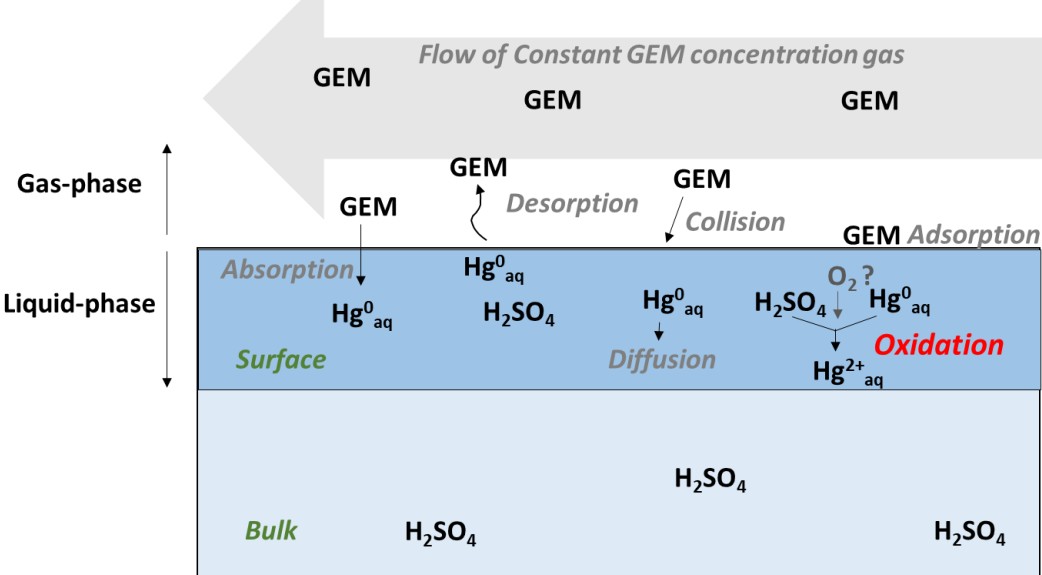

**Figure 5.** A schematic illustration for the oxidation of gaseous elemental mercury in the liquid phase of the acidified water with sulfuric acid.

The experimental uptake coefficient $\gamma_{\text{meas}}$ (the total number of atoms of $THg_{aq}$ found in the solution divided by the total number of GEM collisions at the solution's surface) for the dilute sulfuric acid solutions was determined by the following equation:

$$\gamma_{\text{meas}} = \frac{N_{THgaq}}{\frac{Ac N_{GEM}}{4}}$$

where $N_{THgaq}$, $A$, $\hat{c}$, and $N_{GEM}$ are the total number of $TH_{aq}$ atoms in the solution, the surface area of the solution, the mean atomic speed of GEM and the number density of GEM atoms in the gas phase, respectively. The denominator of the equation indicates the total number of collisions of GEM atoms at the solution's surface. Using this equation, the average SD of the estimated uptake coefficients was $(5.5 \pm 1.6) \times 10^{-6}$. Given a 10 μm diameter for the acid water droplets, a droplet density of $10^3$ cm$^{-3}$, a GEM concentration of 2 ng m$^{-3}$ and a temperature of 293 K, the sinking rate of GEM by the heterogeneous

uptake of acid atmospheric droplets was approximately 4970 years. Compared to the one-year atmospheric lifetime of GEM reacting with an OH radical, this sinking rate is negligibly small. However, it should be noted that, in this calculation, the diffusivity of GEM in the gas phase was ignored, because the standard gas was always replenished. It should also be noted that the calculated atmospheric lifetime has not yet accounted for the mass accommodation of GEM on the surface of the solution, as well as variability of the droplet size and its number density. Depending on the circumstances, those parameters and variables may significantly shorten the lifetime of GEM.

## 4. Conclusions

In this study, the heterogeneous uptake of GEM by pure water, seawater and acidified water with sulfuric and nitric acid within the reported pH range of acid rain was investigated for the first time. The results demonstrated that the uptake of GEM was significantly enhanced for the solutions acidified with sulfuric acid. However, the estimated uptake coefficient was $(5.5 \pm 1.6) \times 10^{-6}$, and under typical atmospheric conditions, the atmospheric lifetime of GEM by this sinking pathway was 4970 years, negligibly small compared with other major sinking pathways. However, this value may be highly variable, depending on the mass accommodation coefficient of GEM, droplet size and number concentration.

**Funding:** This project was financially supported by the internal competitive funding of the National Institute for Minamata Disease (RS17-17, RS18-17, RS19-17, and RS-20-11).

**Institutional Review Board Statement:** Not applicable.

**Informed Consent Statement:** Not applicable.

**Data Availability Statement:** The data presented in this study are available in this manuscript.

**Acknowledgments:** The author acknowledges Nanami Yamamoto for her assistance in this project. The author also acknowledges Kohji Marumoto for his advice on the analysis of $Hg^0_{aq}$ at the beginning of this project.

**Conflicts of Interest:** The authors declare no conflict of interest.

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
