# Peer review of "Oxidation of Gaseous Elemental Mercury in Acidified Water: Evaluation of Possible Sinking Pathway of Atmospheric Gaseous Mercury in Acid Cloud, Fog, and Rain Droplets"

_applsci, doi:10.3390/app11031196_

Round 1

Reviewer 1 Report

This is a nice manuscript that investigates the possible pathway for Hg removal in acidic water environment. Although experiment is well performed, some results are not well presented and the discussion must be improved.

I find some abbreviations very strange. DGM is used, although dimethylmercury is definitely not the part of the fraction. DOM is also very inappropriate as it is usually used for 'dissolved organic matter'. dHg(II) or similar would be more appropriate.

The section about rate determination is welcomed, as gives novelty to the paper, but it has missing words. Also calculations cannot be verified (no graphs), and the Equation used for the calculation might be wrong.

Specific comments are given in attached file.

(Please include line numbers to revision.)

Reviewer 2 Report

In this paper author has tried to explain possible pathway for the atmospheric mercury sinking and dependence of that sinking on the pH value of water present in the atmosphere (as part of the clouds/fog and in rain).

I have found some issues within the text that it should be necessary for evaluating. The author claims in several occasions that the difference between room pressure and the pressure inside the glass vessel is less than 1.3 Pa. It was not, however, obvious, how that pressure was measured and sinci it is very small difference what is the significance of even stating that number. I believe that during the experiments atmospheric pressure fluctuated much more then 1.3 Pa; the possible difference was probably somewhere in the range of kPa.

The author also claims that the experiment was carried out at "near standard temperature" but then claims that the temperature was between 293 and 297 K which is 20+ K far from the standard conditions. Furthermore, later in the text it was claimed that the temperature was kept constant at 323 K. So it has to be clarified.

The author claims that, based on the standard deviation, the detection limits are 30 pg/L (DGM), 150 pg/L (DOM), 9 pg and 44 pg. However, later in the table, measured values for DGM are almost all below the 30 pg/L. I am a little bit confused with that data.

All-in-all if the author can explain the questions raised, this paper can be considered for publication.

Reviewer 3 Report

Abstract

  1. Why did the author choose a “50-100” times higher concentration than the typical atmospheric concentration?
  2. “but lower than the equilibrated DGM … the gas-phase and solution”. I think the author is talking about observed DGM is lower than the equilibrated DGM but this half sentence is sharing the same subject as the previous half sentence.
  3. “The evidence for the enhancement … where strong acid, rain, fog and cloud droplets are formed and stayed in the atmosphere”. Please refer to comment 12.

Introduction

  1. “They concluded that the high concentrations were stability of …” Please rephrase this sentence to make is understandable.

Materials and Method

  1. “The seawater was purged for two days with …”. Did the author measure the DGM and DOM after the purging? What is the purpose of the purging?
  2. Section 2.2. Based on the description of this section, there is no guarantee that DGM can be fully removed through the purging before adding tin chloride solution. Did the author take this into account when calculating the DOM concentration?

Results and Discussion

  1. Also, the recovery yield (Table 1) is not 100%. Did the author take this into account when estimating DGM and DOM? Lastly, why the recovery yield is higher when increasing spkied Hg2+? Please combine with comment 6 and provide your answer.
  2. Section 3.3. “The weight measurement of the vessel with the solution … the determination of DGM and DOM concentrations.” and “It should be noted that the weight measurements of the glass vessel with the uptake … was neglected in the following discussion” are repetitive information.
  3. “DOM concentrations in the seawater were lower than DGM … were further lower”. Please rephrase this sentence to make it understandable.
  4. “The difference in pH between the nitric acid and sulfuric acid … for the sulfuric and nitric acid solutions, respectively.” How could two times difference of hydronium ions lead to a huge difference of DOM in sulfuric and nitric acid solutions?
  5. Figure 2(2). I suggest the author mark the estimated DGM concentration using the Henry’s Law on this figure to demonstrate that the observed DGM is lower than what predicted using Henry’s law.
  6. “Given pH is 3.4 … Compared to the oxidation of DGM by dissolved ozone in the bulk phase ... by the oxidation with OH radical, 6 months or a year.” Please modify and rephrase this sentence. Some information is missing. Also, if the reaction of DGM in sulfuric acid solution is 97 times slower than the reaction with ozone, why it is still a “non-negligible pathway of atmospheric GEM”?

Round 2

Reviewer 1 Report

Dear Author,

the paper is now much improved. It is much easier to read and has novel findings (retention time).

I only have two comments. First, data in Table 2 are hard to read (first two rows).

The second issue are the Henry's constants from Clever et al. They list several values, so I guess you probably took average. If so, please add "approximately" before numbers. Also, I would prefer constants in units bar, not atmosphere, as atmosphere is not an SI unit.

Thanks for this nice paper.

Reviewer 2 Report

I would like to thank the author for his reply and I am now satisfied with this article. Therefore, I suggest to the Editor to publish this article in its present form.

Reviewer 3 Report

Thank you for the response. The paper has been significantly improved. Please provides figures with high resolution. Now the figures are barely discernible. 
